# Automatic Program Synthesis of Long Programs with a Learned Garbage Collector

Amit Zohar[1]                      Lior Wolf [1] [2]

[1]The School of Computer Science , Tel Aviv University
[2]Facebook AI Research

## Abstract

We consider the problem of generating automatic code given sample input-output pairs. We train a neural network to map from the current state and the outputs to the program's next statement. The neural network optimizes multiple tasks concurrently: the next operation out of a set of high level commands, the operands of the next statement, and which variables can be dropped from memory. Using our method we are able to create programs that are more than twice as long as existing state-of-the-art solutions, while improving the success rate for comparable lengths, and cutting the run-time by two orders of magnitude. Our code, including an implementation of various literature baselines, is publicly available at `https://github.com/amitz25/PCCoder`

## 1  Introduction

Automatic program synthesis has been repeatedly identified as a key goal of AI research. However, despite decades of interest, this goal is still largely unrealized. In this work, we study program synthesis in a Domain Specific Language (DSL), following [1]. This allows us to focus on high-level programs, in which complex methods are implemented in a few lines of code. Our input is a handful of input/output examples, and the desired output is a program that agrees with all of these examples.

In [1], the authors train a neural network to predict the existence of functions in the program and then employ a highly optimized search in order to find a correct solution based on this prediction. While this approach works for short programs, the number of possible solutions grows exponentially with program length, making it infeasible to identify the solution based on global properties.

Our work employs a step-wise approach to the program synthesis problem. Given the current state, our neural network directly predicts the next statement, including both the function (operator) and parameters (operands). We then perform a beam search based on the network's predictions, reapplying the neural network at each step. The more accurate the neural network, the less programs one needs to include in the search before identifying the correct solution.

Since the number of variables increases with the program's length, some of the variables in memory need to be discarded. We therefore train a second network to predict the variables that are to be discarded. Training the new network does not only enable us to solve more complex problems, but also serves as an auxiliary task that improves the generalization of the statement prediction network.

Our approach is relatively slow in the number of evaluations per time unit. However, it turns out to be much more efficient with respect to selecting promising search directions, and we present empirical results that demonstrate that: (i) For the length and level of accuracy reported in [1], our method is two orders of magnitude faster. (ii) For the level of accuracy and time budget used in that paper, we are able to solve programs more than two times as complex, as measured by program length (iii) For the time budget and length tested in the previous work, we are able to achieve a near perfect accuracy.

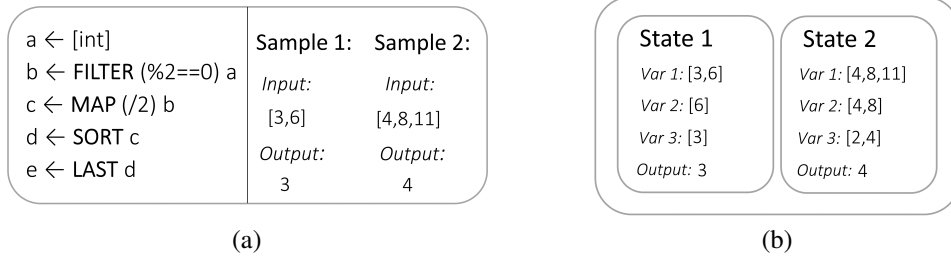

Figure 1: (a) An example of a program of 4 statements in the DSL that receives a single int array as input, and 2 input-output samples. (b) The environment of the program after executing line 3.

## 2 General Formulation

Our programs are represented by the Domain Specific Language (DSL) defined by [1]. Each program is a sequence of function calls with their parameters, and each function call creates a new variable containing the return value. The parameters are either inputs, previously computed variables, or a lambda function. There are 19 lambda functions defined in this DSL, such as increasing every element of the list by one (+1), thresholding (>0) or parity checks (%2==0), and are used, for example, as inputs to the MAP operator that applies these functions to each element in a list.

Each variable in the DSL is either an integer in the range of [-256,255] or an array of integers, which is bounded by the length of 20. The output of the program is defined as the return value of its last statement. See Fig. 1(a) for a sample program and appendix F of [1] for the DSL specification.

Formally, consider the set of all programs with length of at most $t$ and number of inputs of at most $n$. Let $\mathcal{F}$ be the set of all operators (functions) in our DSL and let $\mathcal{L}$ be the set of all lambdas. The set of values for each operand is $\mathcal{M} = \{i \in \mathbb{N} \mid 1 \leq i \leq t + n\} \cup \mathcal{L}$. For an operator $o$ with $q$ operands, the statement would include the operator and a list of operands $p = [p_1, ..., p_q] \in \mathcal{M}^q$. Many of the elements of $M^q$ are invalid if a variable $p_j$ was not initialized yet, or if the value $p_j$ is not of the suitable type for operand $j$ of the operator $o$. We denote the set of possible statements in a program by $\mathcal{S} \subseteq \mathcal{F} \times M^r$, where $r$ is the maximal number of operands for an operator in the DSL. $\mathcal{S}$ contains both valid and invalid statements, and is finite and fixed. The network we train learns to identify the valid subset, and the constraints are not dictated apriori, e.g., by means of type checking.

In imperative programming, the program advances by changing its state one statement at a time. We wish to represent these states during the execution. Naturally, these states depend on the input. For convenience, we also include the output in the state, since it is also an input of our network. To that end we define the **state** of a program as the sequence of all the variable values acquired thus far, starting with the program's input, and concatenated with the desired output. Before performing any line of code, the state contains the inputs and the output. After each line of code, a new variable is added to the state.

A single input/output pair is insufficient for defining a program, since many (inequivalent) potential programs may be compatible with the pair. Therefore, following [1] the synthesis problem is defined by a set of input/output pairs. Since each input-output example results in a different program state, we define the **environment** $e$ of a program at a certain step of its execution, as the concatenation of all its states (one for each input/output example), see Fig. 1(b). Our objective is thus to learn a function $f : \mathcal{E} \rightarrow \mathcal{S}$ that maps from the elements of the space of program environments $\mathcal{E}$ to the next statement to perform.

By limiting the program's length to $t$, we are able to directly represent the mapping $f$ by a feed forward neural network. However, this comes at a cost of limiting the program's length by a fixed maximal length. To remedy this, we learn a second function $g : \mathcal{E} \rightarrow \{0, 1\}^{t+n}$ that predicts for each variable in the program whether it can be dropped. During the program generation, we can use this information to forgo variables that are no longer needed, thereby making space for new variables. This enables us to generate programs longer than $t$.

Note that our problem formulation considers the identification of any program that is compatible with all samples to be a success. This is in contrast to some recent work [2, 3, 4], focused on

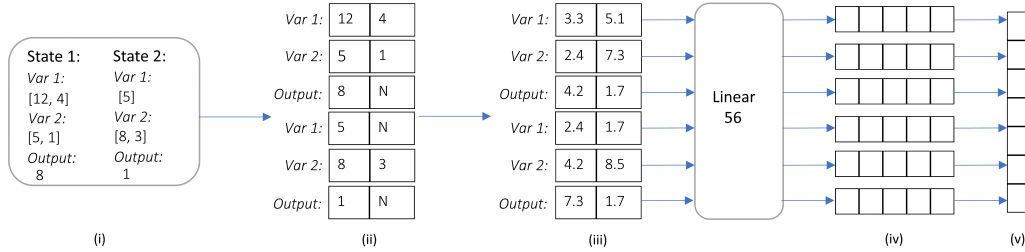

Figure 2: A depiction of the embedding module of our network. In this example there are $n = 2$ examples, $v = 2$ variables, arrays are of maximum length $l = 2$ and the embedding dimension is $d = 1$. For simplicity, the variable type one-hot encoding is omitted. (i) A program environment is given as input to the network. (ii) Each variable is represented by a compact fixed-length vector. N stands for NULL. (iii) The integers of each variable are embedded using a learned look-up table. (iv) Each vector is passed through a 56-unit linear layer. (v) The resulting vectors are concatenated to a single fixed-length vector.

string processing, where success is measured on unseen samples. Since any string can be converted to any other string by simple insertions, deletions, and replacements, string processing requires additional constraints. However, in the context of our DSL, finding any solution is challenging by itself. Another difference from string processing is that strings can be broken into segments that are dealt separately [5].

Within the context of string processing, [6] is perhaps the closest work to ours. Their approach is to treat the problem as a sequence-to-sequence problem, from samples to code. As part of our ablation analysis we compare our method with this method. Our setting also differs from frameworks that complete partial programs, where parts of the program are provided or otherwise constrained [7, 8].

## 3 Method

We generate our train and test data similarly to [1]. First, we generate random programs from the DSL. Then, we prune away programs that contain redundant variables, and programs for which an equivalent program exists in the dataset (could be shorter). Equivalence is approximated by identical behavior on a set of input-output examples. We generate valid inputs for the programs by bounding their output value to our DSL's predetermined range and then propagating these constraints backward through the program.

### 3.1 Representing the Environment

The environment is the input to both networks $f$ and $g$ and is represented as a set of $k$ state-vectors, where $k$ is the number of samples. Each state vector is represented by a fixed-length array of $v$ variables, which includes both generated variables and the program's initial input, which is assumed to contain at least one variable and up to $n$ variables. If there are less than $v$ variables, a special NULL value appears in the missing values of the array.

Each variable is first represented as a fixed-length vector, similarly to what is done in [1]. The vector contains two bits for a one-hot encoding of the type of the variable (number or list), an embedding vector of size $d = 20$ to capture the numeric value of the number or the first element of the list, and $l - 1$ additional cells of length $d$ for the values of the other list elements. In our implementation, following the DSL definition of [1], $l = 20$ is the maximal array size. Note that the embedding of a number variable and a list of length one differs only in the one-hot type encoding.

The embedding of length $d$ is learned as part of the training process using a Look Up Table that maps integers from $[-256, 256]$ to $\mathbb{R}^d$, where 256 indicates a NULL value, or a missing element of the list if its length is shorter than $l$.

A single variable is thus represented, both in our work and in [1], by a vector of length $l \cdot d + 2 = 402$. At this point the architectures diverge. In our work, this vector is passed through a single layer neural network with 56 units and a SELU activation function [9]. A state is then represented

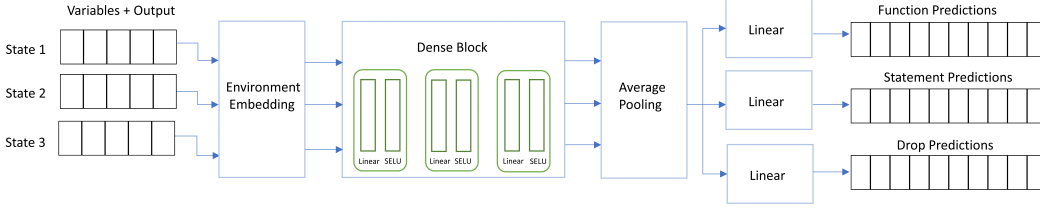

Figure 3: The general architecture of our network. Each state vector is first embedded and then passed through a dense block before finally pooling the results together to a single vector. The result is then passed to 3 parallel feed-forward layers that perform the predictions.

by a concatenation of $v + 1$ activation vectors of length 56, with the additional vector being the representation of the output. Thus, we obtain a state vector of length $s = 56(v + 1)$.

## 3.2 Network Architecture

We compute $f$ and $g$ based on the embedding of the environment, which is a set of $k$ vectors in $\mathbb{R}^s$. Since the order of the examples is arbitrary, this embedding is a permutation-invariant function of these state vectors.

Since we perform a search with our network, an essential desired property for our architecture is to have high model capacity while retaining fast run times. To this end, we use a densely connected block [10], an architecture that has recently achieved state-of-the-art results on image classification with relatively small depth. In our work, the convolution layers are replaced with fully connected layers. Permutation invariance is then obtained by employing average pooling.

Each state vector is thus passed through a single 10-layer fully-connected dense block. The $i$-th layer in the block receives as input the activations of all layers $1, 2, .., i - 1$, and has 56 hidden units, except for the last layer which has 256 units. SELU [9] activations are used between the layers of the dense block. The resulting $k$ vectors are pooled together via simple arithmetic averaging.

In order to learn the environment embedding and to perform prediction, we learn networks for three different classification tasks, out of which only the two matching functions $f$ and $g$ are used during test time. **Statement prediction:** An approximation of $f$ from the formulation in Sec. 2. Since we limit the amount of variables in the program, there is a finite set of possible statements. The problem is therefore cast as a multiclass classification problem. **Variable dropping:** An approximation of $g$ from the formulation. This head is a multi-label binary classifier that predicts for each variable in the program whether it can be dropped. At test-time, when we run out of variables, we drop those that were deemed most probable to be unnecessary by this head. **Operator prediction:** This head predicts the function of the next statement, where a function is defined as the operator in combination with the lambda function in case it expects one, but without the other operands. For example, a map with $(+1)$ is considered a different function from a map with $(-1)$. While the next function information is already contained inside the predicted statement, we found this auxiliary training task to be highly beneficial. Intuitively, this objective is easier to learn than the objective of the first classification network, since the number of classes is much smaller and the distinction between the classes is clearer. In addition, it provides a hierarchical structure to the space of possible statements.

In our implementation, all three classifiers are linear. The size of the output dimensions for the statement prediction and variable dropping are thus $|\mathcal{S}|$ and $[0, 1]^v$ respectively, and the size of the output of the function prediction is bounded by $|\mathcal{F}||\mathcal{L}|$.

During training, for each program in our dataset we execute the program line by line and generate the following information: (i) The program environment achieved by running the program for every example simultaneously, up until the current line of code. (ii) The next statement of the program. (iii) The function of the next statement. (iv) A label for each variable that indicates whether it is used in the rest of the program. If not, it can be dropped.

Cross entropy loss is used for all three tasks, where for variable-dropping multi-label binary cross entropy is used. Concretely, denote the predicted statements, functions and drop probability for variable $i$ by $S, F, D_i$ respectively, and by $\hat{S}, \hat{F}, \hat{D}_i$ the corresponding ground truths. Our model is

trained with the following unweighted loss:

$$L = \text{CE}(S, \hat{S}) + \text{CE}(F, \hat{F}) + \sum_{i=1}^{v} \text{CE}(D_i, \hat{D}_i)$$

For optimization, we use Adam [11] with a learning rate of 0.001 and batch size of 100.

### 3.3  The Search Method

Searching for the program based on the network's output is a tree-search, where the nodes are program environments and the edges are all possible statements, ordered by their likelihood according to the network. During search, we maintain a program environment that represents the program predicted thus far. In each step, we query the network and update the environment according to the predicted statement. If the maximum number of variables $v = t + n$ is exceeded, we drop the variable predicted most likely to be insignificant by the model's drop ("garbage collection") sub-network.

Since with infinite time any underlying program can be reached and since the benchmarks are time-based, we require a search method that is both exhaustive and capable of being halted, yielding intermediate results. To this end, we use CAB [12], a simple extension of beam search that provides these exact properties.

CAB operates in a spiral manner. First, a regular beam search is run with initial heuristic pruning rules. If a desired solution could not be found, the pruning rules are weakened and a new beam search begins. This operation is performed in a loop until a solution is found, or when a specified timeout has been reached. Specifically, the parameters of CAB in our setting are: $\alpha$, which is the beam size; $\beta$, which is the number of statement predictions to examine; and $c$, a constant value that is added to $\beta$ after every beam search. After every iteration, CAB doubles $\alpha$ and $\beta$ is increased by $c$, ensuring that the vast majority of paths explored in the next iteration will be new. We employ $\alpha = 100$, $\beta = 10$, and $c = 10$. It is interesting to note that in [1], a search that is similar in spirit to CAB was attempted in one experiment, but resulted in reduced performance.

## 4  Experiments

We ran a series of experiments testing the required runtime to reach a certain level of success. As baselines we employed the best variant of DeepCoder [1], and the untrained DFS search employed as a baseline in that paper. We employed various variants of our Predict and Collect Coder (PCCoder) method, in order to understand its sensitivity and to find out which factors contribute to its success. All our experiments were performed using F64s Azure cloud instances. Each search is a Cython program that runs on a single thread.

In each line of experiments, we trained our model on a dataset consisting of programs of length of up to $t_1$. We then sampled $w$ test programs of length $t_2$, which are guaranteed to be semantically disjoint from the dataset.

The first line of experiments was done similarly to [1], where our model was trained on programs of length of up to $t_1 = 4$ and tested on $w = 100$ programs of length $t_2 = 5$. For the environment representation, we used a memory buffer that can hold up to four variables in addition to the input. This means that at test-time for programs with three inputs a variable is dropped for the last statement.

Since there is no public implementation for [1], we have measured our dataset on our reimplementation thereof. Differences in results between the article and our reimplementation can be explained by the fact that our code is in optimized Cython, whereas [1] used C++. To verify that our data is consistent with [1] in terms of difficulty, we have also provided results for baseline search of our reimplementation and the article.

The results are reported in the same manner used in [1]. For every problem, defined by a set of five input/output pairs, the method is run with a timeout of 10,000s. We then consider the time that it took to obtain a certain ratio of success (number of solved problems over the number of problems after a certain number of seconds). The results are reported in Tab. 1. As can be seen, our reimplementation of DeepCoder is slower than the original implementation (discarding hardware differences). However, the new method (PCCoder) is considerably faster than the original DeepCoder method, even though our implementation is less optimized.

Table 1: Recreating the experiment of [1], where training is done on programs shorter than 5 statements and testing is done on programs of length 5. The table shows the time required to achieve a solution for a given ratio of the programs. Success ratios that could not be reached in the timeout of 10,000 seconds are marked by blanks.

| Method | 20% | 40% | 60% | 80% | 95% |
|---|---|---|---|---|---|
| Baseline [1] | 163s | 2887s | 6832s | - | - |
| Baseline (reimpl.) | 484s | 4310s | 9628s | - | - |
| DeepCoder [1] | 24s | 514s | 2654s | - | - |
| DeepCoder (reimpl.) | 67s | 1465s | 3826s | - | - |
| PCCoder | 5s | 41s | 259s | 782s | 2793s |

Table 2: Comparison for programs of varying lengths between our $PCCoder_8$ method with memory size 8, our $PCCoder_{11}$ with memory size 11, and DeepCoder (*reimplementation)

| Length | Model | Total solved | Timeout needed to solve | | | | | | | | |
|---|---|---|---|---|---|---|---|---|---|---|---|
| | | | 5% | 10% | 20% | 40% | 60% | 70% | 80% | 90% | 99% |
| 5 | DeepCoder* | 63.0% | 0.6s | 12s | 25s | 458s | 2004s | - | - | - | - |
| | $PCCoder_8$ | 99.2% | 1s | 1s | 2s | 4s | 5s | 13s | 45s | 132s | 1743s |
| | $PCCoder_{11}$ | 99.6% | 1s | 2s | 3s | 4s | 6s | 11s | 37s | 129s | 1587s |
| 8 | DeepCoder* | 11.2% | 578s | 2979s | - | - | - | - | - | - | - |
| | $PCCoder_8$ | 90.0% | 1s | 2s | 4s | 24s | 168s | 454s | 1050s | 4759s | - |
| | $PCCoder_{11}$ | 91.2% | 4s | 5s | 8s | 21s | 127s | 310s | 988s | 4382s | - |
| 10 | DeepCoder* | 7.0% | 642s | - | - | - | - | - | - | - | - |
| | $PCCoder_8$ | 73.0% | 2s | 4s | 6s | 143s | 1642s | 3426s | - | - | - |
| | $PCCoder_{11}$ | 74.0% | 5s | 7s | 12s | 135s | 1299s | 3285s | - | - | - |
| 12 | DeepCoder* | 4.0% | - | - | - | - | - | - | - | - | - |
| | $PCCoder_8$ | 65.4% | 3s | 4s | 5s | 295s | 2725s | - | - | - | - |
| | $PCCoder_{11}$ | 67.2% | 8s | 11s | 21s | 378s | 2589s | - | - | - | - |
| 14 | DeepCoder* | 2.0% | - | - | - | - | - | - | - | - | - |
| | $PCCoder_8$ | 52.0% | 2s | 3s | 39s | 747s | - | - | - | - | - |
| | $PCCoder_{11}$ | 60.4% | 1s | 2s | 18s | 162s | 4476s | - | - | - | - |

In the second experiment, we trained our model on a single dataset consisting of programs of length of up to $t_1 = 12$. We then sampled $w = 500$ programs of lengths 5, 8, 10, 12, 14 and tasked our model with solving them. In total, the dataset consists of 143000 train programs of varying lengths and 2500 test programs. For each test length $t_2$, the model is allowed to output programs of maximum length $t_2$. The memory of the programs was limited to either 11 (including up to three inputs) or 8. For comparison, results of our reimplementation of [1] for the same tests are also provided.

In this experiment, in order to limit the total runtime, we employed a timeout of 5000s. The results are reported in Tab. 2. As can be seen, for programs of length 5, when trained on longer programs than what was done in the first experiment, our method outperforms DeepCoder with an even larger gap in performance.

Overall, within the timeout boundary of 5000s, our method, with a memory of size 11, solves 60% of the programs of length 14, and 67% of those of length 12. DeepCoder, meanwhile, fails to surpass the 5% mark for these lengths. The results for memory size 11 and memory size 8 are similar, except for length 14. Considering that with three inputs, the first memory would need to discard variables after step 8, both variants make an extensive use of the garbage collection mechanism.

Tab. 3 compares the length of the predicted program with the length of the ground-truth. As can be seen, for lengths 5 and 8 most predictions are equal in length to the solutions, whereas for larger lengths it is common to observe shorter solutions than the target program.

Table 3: The length of the predicted program (columns) vs. corresponding ground-truth length (rows).

| | Length | 1 | 2 | 3 | 4 | 5 | 6 | 7 | 8 | 9 | 10 | 11 | 12 | 13 | 14 |
|---|---|---|---|---|---|---|---|---|---|---|---|---|---|---|---|
| $PCCoder_8$ | 5 | 0% | 0% | 1% | 14% | 85% | - | - | - | - | - | - | - | - | |
| | 8 | 0% | 0% | 0% | 1% | 10% | 32% | 23% | 34% | - | - | - | - | - | - |
| | 10 | 0% | 0% | 0% | 2% | 5% | 16% | 16% | 32% | 16% | 13% | - | - | - | - |
| | 12 | 0% | 0% | 0% | 3% | 3% | 9% | 27% | 22% | 11% | 9% | 8% | 8% | - | - |
| | 14 | 0% | 0% | 0% | 0% | 1% | 5% | 10% | 26% | 20% | 12% | 7% | 7% | 4% | 8% |
| $PCCoder_{11}$ | 5 | 0% | 0% | 2% | 16% | 82% | - | - | - | - | - | - | - | - | |
| | 8 | 0% | 0% | 1% | 1% | 8% | 25% | 29% | 36% | - | - | - | - | - | - |
| | 10 | 0% | 0% | 0% | 2% | 5% | 12% | 20% | 26% | 19% | 16% | - | - | - | - |
| | 12 | 0% | 0% | 0% | 2% | 2% | 9% | 19% | 21% | 14% | 15% | 10% | 8% | - | - |
| | 14 | 0% | 0% | 0% | 0% | 1% | 9% | 7% | 23% | 17% | 13% | 19% | 7% | 1% | 3% |

Table 4: CIDEr scores of PCCoder's predictions with respect to the ground-truth. Each program is represented as a sentence, and each function is a word. Only successful predictions are measured.

| | 5 | 8 | 10 | 12 | 14 |
|---|---|---|---|---|---|
| $PCCoder_8$ | 72.87 | 48.47 | 33.11 | 27.26 | 19.66 |
| $PCCoder_{11}$ | 72.33 | 52.93 | 38.48 | 28.41 | 21.90 |

Following this, one may wonder what is the amount of similarity between the target program and the found one, when the search is successful. We employ the CIDEr [13] score in order to compute this similarity. CIDEr is considered a relatively semantic measure for similarity between two sentences, and is computed based on a weighted combination of n-grams that takes into account the words' frequency. In our experiment, we consider each program to be a sentence and each valid combination of functions and lambdas to be a word. We use the training set of programs of length of up to $t_1 = 12$ to compute the background frequencies. The results are presented in Tab. 4. As can be seen, for length 5 the similarity is relatively high, whereas for larger lengths the score gradually decreases.

We next perform an ablation analysis in order to study the contribution of each component. In this experiment, we trained different models on a dataset of 79000 programs of length of up to $t_1 = 8$. We then tested their performance on 1000 test programs of length $t_2 = 8$. All variants employ a memory size of $v = 8$, which means that variables are discarded after step five for programs with 3 inputs. Specifically, these are the variants tested: *PCCoder* - the original PCCoder model. *PCCoder_SD* - the PCCoder model with the auxiliary function task discarded (called SD since statement and variable drop predictions are still active). *PCCoder_SF* - the PCCoder model with the variable dropping task discarded. The variables are discarded at random. *PCCoder_S* - the PCCoder model with both the variable dropping and function prediction tasks discarded. *PCCoder_FixedGC* - the PCCoder model where the GC is learned after the representation is fixed. *PCCoder_ImportGC* - the PCCoder model where GC is not used as an auxiliary task, but computed separately using the original network. *PCCoder_Linear* - the PCCoder model with the dense block replaced by 3 feed-forward layers of 256 units. This results in a similar architecture to [1], except for differences in input and embedding. Note that training deeper networks, without employing dense connections failed to improve the results. *PCCoder_ResNet* - the PCCoder model with ResNet blocks instead of the dense block. We found that using 4 ResNet blocks, each comprised of 2 fully-connected layers with 256 units and a SELU activation worked best. *PCCoder_DFS* - the PCCoder model, but with DFS search instead of CAB. The width of the search is limited to ensure that we try multiple predictions for the lower depths of the search. *PCCoder_Ten* - the original PCCoder model trained on the same dataset but with 10 examples for each program (both at train and test time). *PCCoder_Ten5* - the results of *PCCoder_Ten* where at run time we provide only 5 samples instead of 10.

The results are reported in Tab. 5. As can be seen, all variants except PCCoder_Ten5 result in a loss of performance. The loss of the variable drop is less detrimental than the drop of the auxiliary function prediction task, pointing to difficulty of training to predict specific statements. The GC component has a positive effect both as an auxiliary task and on test-time results. Linear and ResNet, which

Table 5: Comparison between several variants of our method. Training was done on programs of length of up to 8 and tested on 1,000 programs of length 8, with a timeout of 1,000 seconds.

| Model | Total solved | Timeout needed to solve | | | | |
|---|---|---|---|---|---|---|
| | | 20% | 40% | 60% | 70% | 80% |
| PCCoder | 83% | 3s | 10s | 84s | 202s | 635s |
| PCCoder_SD | 70% | 5s | 66s | 347s | 953s | |
| PCCoder_SF | 77% | 6s | 11s | 161s | 741s | - |
| PCCoder_S | 66% | 7s | 126s | 660s | - | - |
| PCCoder_FixedGC | 79% | 4s | 13s | 116s | 285s | - |
| PCCoder_ImportGC | 80% | 3s | 12s | 96s | 263s | 939s |
| PCCoder_Linear | 73% | 0.7s | 13s | 323s | 643s | - |
| PCCoder_ResNet | 76% | 2s | 9s | 121s | 414s | - |
| PCCoder_DFS | 67% | 29s | 310s | 692s | - | - |
| PCCoder_Ten | 78% | 1s | 12s | 78s | 229s | - |
| PCCoder_Ten5 | 84% | 0.6s | 9s | 70s | 135s | 530s |

Table 6: Comparison between multiple recently proposed methods of program synthesis on our DSL. Similarly to the ablation experiment, training was done on programs of length of up to 8 and tested on 1,000 programs of length 8, with a timeout of 1,000 seconds.

| Model | Total solved | Timeout needed to solve | | | |
|---|---|---|---|---|---|
| | | 1% | 2% | 4% | 5% |
| PCCoder | 83% | 0.2s | 0.3s | 0.4s | 0.6s |
| PCCoder_No_IO | 4% | 50s | 51s | 554s | - |
| DeepCoder | 5% | 44s | 245s | 311s | 971s |
| DeepCoder_CAB | 4% | 3s | 131s | 626s | - |
| RobustFill (Attn A) | 2% | 8s | 843s | - | - |
| RobustFill (Attn B) | 4% | 7s | 11s | 517s | - |

employ a shallower network, are faster for easy programs, but are, however, challenged by more difficult ones. The beam search is beneficial and CAB is considerably more efficient than DFS.

It has been argued in [14] that learning from only a few sample input/output pairs is hard and one should use more samples. Clearly, training on more samples cannot hurt as they can always be ignored. However, the less examples one uses during test-time, the more freedom the algorithm has to come up with compatible solutions, making the synthesis problem easier. Our experiment with PCCoder_Ten shows that, for a network trained and tested on 10 samples vs a network trained and employed on 5 samples, a higher success rate is achieved with less input samples. However, as indicated by PCCoder_Ten5, if more samples are used during training but only 5 samples are employed at test-time, the success rate of the model improves.

In the next experiment, we evaluate a few other program synthesis methods on our DSL. To this end, we employ the same experimental conditions as in the ablation experiment. These are the methods assessed: (i) Two of the variants of RobustFill [6], reimplemented for our DSL. For fairness, CAB is used. (ii) The original DeepCoder model and a variant of DeepCoder where CAB search is used instead of DFS. (iii) A variant of our PCCoder model where intermediate variable values are not used, causing the state to be fixed per input-output pair.

As can be seen in Tab. 6, using CAB with DeepCoder slightly degrades results in comparison to DFS. A possible reason is that the method's predicted probabilities do not depend on previous statements. Furthermore, the Attn.B variant of RobustFill and the variant of PCCoder without IO states both perform comparably to DeepCoder. Considering that the programs being generated are reasonably long and that all three methods do not track variable values during execution, this can be expected.

A specific program can be the target of many random environments. An auxiliary experiment is performed in order to evaluate the invariance of the learned representation to the specific environment used. We sampled eight different programs of length 12 and created ten random environments for

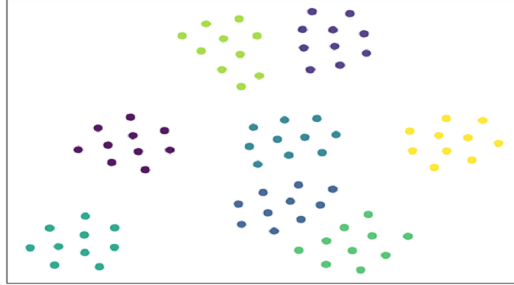

Figure 4: A 2D t-SNE plot of the embedding of the environments in $\mathbb{R}^{256}$ (after average pooling over the samples). The experiment consists of eight different programs of length 12. For each program, ten random environments were created, each including five input-output samples. The eight colors represent the associated programs of each environment.

each, with each environment comprising of five samples of input/output pairs. As can be seen in Fig. 4, the environments for each program are grouped together.

## 5 Conclusions

We present a new algorithm that achieves a sizable improvement over the recent baseline, which is a strong method that was compared to other existing approaches from the automatic code synthesis literature. It would be interesting to compare the performance of the method to human programmers who can reason, implement, and execute. We hypothesize that, within this restricted domain of code synthesis, the task would be extremely challenging for humans, even for programs of shorter lengths.

Since any valid solution is a correct output, and since we employ a search process, one may wonder whether Reinforcement Learning (RL) would be suitable for the problem. We have attempted to apply a few RL methods as well as to incorporate elements from the RL literature in our work. These attempts are detailed in the supplementary.

**Acknowledgments**

This work was supported by an ICRC grant.

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
