[Supplementary Material · supplementary.pdf]

# Automatic Program Synthesis of Long Programs with a Learned Garbage Collector

**Amit Zohar**[1]  **Lior Wolf**[1][2]

[1]The School of Computer Science , Tel Aviv University
[2]Facebook AI Research

## 1 Some Attempts to Apply Reinforcement Learning

A prominent limitation of our method is the reliance, during training, on a supervision in the form of an existing solution. First, the solution does not need to be unique, and many input/output pairs have multiple valid programs that map the input to the output. Training to match one correct solution is probably suboptimal. Second, during test time, there is a search process from a current state to the end goal. It would be helpful if a method could optimize a success score during search, thereby continuing its learning to better solve the sample at hand.

An attractive alternative to supervised learning would be Reinforcement Learning (RL). We have attempted to solve the problem using multiple RL and RL-inspired approaches: (i) an AlphaGo [1] inspired approach; (ii) an imitation learning based approach, following [2]; and (iii) a more modest application of RL, in which we use self-critical training to optimize with respect to an organic scoring mechanism [3].

In the first approach, we initialized our policy network with supervised training and then applied further actor-critic training. Specifically, we used the A3C algorithm ([4]), which is considered amongst the state-of-the-art approaches in most RL settings. However, we found the model extremely unstable under these circumstances.

In the second approach, we treated the ground-truth program statements as expert behavior and attempted to imitate them with the algorithm from [2]. We also tried to apply this form of training on a supervised-learned model. In the article's formulation, this corresponds to initializing policy parameters with behavioral cloning, which should significantly increase learning speed. We hypothesize this approach failed because, while the ground-truth programs are correct solutions, they were generated randomly and thus do not represent any specific policy. Therefore, it is unreasonable to expect an approach that assumes the existence of an underlying policy to succeed.

In the third approach, we attempted several scoring mechanisms. One idea was, given a program state, to search for a correct solution from it. The score would then be the number of steps needed to reach termination. By optimizing with respect to this score, our model indirectly learns its real goal. Other scores were also tried, including a CIDEr variant for program similarity based on functions.

## 2 Example Programs

To illustrate the difficulty of the synthesis, we have attached a few long programs together with their I/O samples from our test dataset, that our model predicts successfully.

## Program 1

```
v0 ← [int]
v1 ← [int]
v2 ← TAIL v1
v3 ← TAIL v0
v4 ← DROP v3 v1
v5 ← MAX v4
v6 ← ACCESS v2 v4
v7 ← DROP v6 v4
v8 ← HEAD v7
v9 ← DROP v5 v7
v10 ← SORT v9
v11 ← MAX v10
v12 ← TAKE v11 v10
v13 ← ACCESS v8 v12
```

| Input | Output |
|---|---|
| [0], [1, 1, 0, 2, 0] | 2 |
| [9, 12, 2, 0, 7, 8, 1, 3, 8, 4, 7, 4, 12, 2, 0], [0, 2, 1, 0] | 0 |
| [0], [0, 7, 6, 0, 1, 5, 4, 3, 3] | 3 |
| [14, 4, 3, 1, 7, 7, 4, 6, 7, 0, 9, 13, 8, 10, 0, 3], [4, 3, 0, 1, 2, 0, 2] | 2 |
| [0, 0, 0, 7, 5, 1, 7, 0, 0], [0, 0, 9, 6, 6, 5, 8, 9, 2, 8, 7, 9, 9, 1] | 1 |

## Program 2

```
v0 ← [list]
v1 ← TAIL v0
v2 ← ACCESS v1 v0
v3 ← DROP v2 v0
v4 ← SORT v3
v5 ← DROP v2 v4
v6 ← FILTER (>0) v5
v7 ← REVERSE v6
v8 ← SORT v7
v9 ← MAP (+1) v8
v10 ← HEAD v9
v11 ← ACCESS v10 v9
v12 ← TAKE v11 v9
```

| Input | Output |
|---|---|
| [4, 2, 2, 0, 4, 3, 3] | [3, 3, 4, 4] |
| [6, 2, 6, 2, 6, 5, 3, 2, 4, 7] | [4, 5, 6, 7, 7, 8] |
| [1, 8, 0, 8, 3, 4, 5, 4, 2, 8, 2] | [2, 3, 3] |
| [3, 1, 3, 0, 5, 4, 6, 6, 3] | [2, 4, 4, 4] |
| [1, 3, 0, 0, 0, 4, 0, 4] | [2, 4, 5, 5] |

## Program 3

```
v0 ← [list]
v1 ← [list]
v2 ← MAX v1
v3 ← MAP (+1) v0
v4 ← REVERSE v3
v5 ← ZIPWITH (+) v4 v4
v6 ← FILTER (>0) v5
v7 ← REVERSE v6
v8 ← MIN v7
v9 ← TAKE v8 v7
v10 ← SORT v9
v11 ← REVERSE v10
v12 ← SCAN1L (+) v11
v13 ← MAX v12
v14 ← TAKE v2 v12
v15 ← TAKE v13 v14
```

| Input | Output |
|---|---|
| [2, 1, -1, 2, 4, -1, -1, 3, 1, 4, 4, 4, -1, 2, 0, 5], [57, 133, 220, 231, 186, 82, 45, 14, 227, 227, 89, 109] | [6, 10] |
| [1, 0, 7, -1, 1, 3, 7, 2, 5, 7, -1, 1, 4, 1], [188, 237, 212, 202, 50, 19, 232] | [4, 6] |
| [17, 13, -1, 0, 8], [253, 51] | [36, 64] |
| [6, 5, 0, 6, 3, 4, 1, 7, 7, 7, 3, 8], [42, 59, 64, 29, 186, 102, 186, 141] | [14, 26] |
| [12, 3, 8, 8, 12, 6, 11, 2], [246, 113, 222, 18, 144, 250, 6, 63] | [26, 52, 70, 88, 102, 110] |

## Program 4

```
v0 ← [list]
v1 ← [int]
v2 ← TAIL v0
v3 ← ACCESS v2 v0
v4 ← TAKE v1 v0
v5 ← FILTER (>0) v4
v6 ← TAIL v5
v7 ← ACCESS v6 v5
v8 ← DROP v3 v5
v9 ← MAX v8
v10 ← TAKE v9 v8
v11 ← REVERSE v10
v12 ← SUM v11
v13 ← TAKE v7 v11
v14 ← TAKE v12 v13
v15 ← MAP (+1) v14
```

| Input | Output |
|---|---|
| [0, 2, 2, 3, 3], 238 | [4] |
| [10, 8, 6, 2, 8, 0, 10, 3, 6, 0, 3, 1, 2], 197 | [3, 2, 4, 7, 4] |
| [2, 1, 1, 1], 125 | [2] |
| [4, 8, 10, 13, 6, 5, 1, 8, 1, 5, 2, 10, 13, 4, 6, 9, 3, 4], 11 | [3, 6, 2, 9, 2] |
| [0, 1, 1], 72 | [2] |

# 3 Impact of DSL instructions

To assess which parts of the DSL are more problematic, we have sorted the functions by the frequency in failing experiments with the function normalized by its overall frequency. As might be expected, the "easiest" functions are: TAKE, SUM, COUNT, and the "hardest" are DROP, REVERSE, MAP.