[Reviews · NeurIPS 2018]

Reviewer 1



The paper presents a modification of a previous baseline method in the application area of inductive program synthesis (IPS). This new architecture, PCCoder, shows some considerable improvement over DeepCoder, the baseline main method it compared against. Within the setting of IPS, the goal of the model is, provided with some input/output examples meant to represent a partial program specification, output a program that successfully maps from input to output for each I/O example in the set. A program is considered correct if it accomplishes the mapping on all provided I/O pairs (there is no notion of generalization to unseen I/O pairs as sometimes done in previous work, especially in string processing domains). The input/output pairs consist of integers, and the program is constructed from the relatively expressive Domain Specific Language defined in the DeepCoder paper. The model maintains a program state as it iteratively chooses which functions to execute on the input examples, and this state contains a set of variables that can be used to store partial results. In contrast to DeepCoder, PCCoder additionally has a garbage collector that can delete unused variables. The paper is very clearly written and easy to follow. I found that most details of the architecture and the method used to train it can be clearly understood from the text. They also released their code with the paper, so that their specific implementation details are provided. The architecture itself is closely related to DeepCoder, but contains some noticeable differences in activations used (SeLU v.s. sigmoid in I/O embedding function) and using a deep network (10-layer) of dense blocks to process the program state, among other differences. It also includes a novel garbage collection operation that allows variable deletion (potentially enabling a much larger class of programs to be generated) and used a beam-search-like method called CAB in place of the DFS, enumeration, sketch and lambda^2 search techniques used in DeepCoder. The experimental results section shows some considerable improvement over the DeepCoder baseline, including on program lengths much longer than originally reported in the DeepCoder paper (up to 14, while DeepCoder only evaluated on program lengths up to 5). The results seem impressive, with a 2-3 order of magnitude speedup of DeepCoder shown in Table 2. Due to the lack of an open-source implementation of DeepCoder, the authors had to reimplement it but obtained slightly worse results due to less optimized code. Despite the poorer performance, they do provide results that compare their reimplementation with the results in the original paper side-by-side in Table 1. A relatively comprehensive ablation analysis reveals that no single architectural change from DeepCoder is responsible for all the improvement in performance. An interesting thing to observe would be to run DeepCoder with the improved search method of CAB to see if it provides any significant improvement (since in the ablation analysis, using DFS with PCCoder caused a large decrease in performance). In summary, the paper is well-written and the results seem to show a clear improvement over a previous deep-network-based IPS baseline. Some weaknesses are that the paper’s results are largely engineering-related, as the architecture itself does not contain many novel components but combines previous work (dense layers, SeLU, CAB) into a higher performing model. Despite this, the results show a definite improvement over the baseline and extend deep IPS models to work on longer program lengths.

Reviewer 2



= Summary The paper presents a program synthesis framework interleaving execution of partially generated programs with a learned component to direct program completion. To combat the growing number of initialized variables, a second component learning to drop ("garbage collect") unneeded variables is learned. Overall, the paper presents a well-executed evolution of the DeepCoder idea; rather incremental results are balanced by significant experimental evidence for their value. = Originality The core idea of learning to predict the most likely next statement based on (current program state, output) tuples (instead of (initial state, output) tuples) is rather obvious. It's present in the reviews for the DeepCoder paper (cf. point 2 of https://openreview.net/forum?id=ByldLrqlx¬eId=HJor8rbNg), in [2] (modulo unclear writing) and also in several concurrent NIPS submissions. This does not mean that the idea is without merit; but that on its own, it does not contribute a significant contribution. The second idea, of learning which variables to ignore, seems original and is interesting. = Clarity The paper is reasonably well-written, but could profit from a few cleanups. For example, Fig. 2 and Fig. 3 could be integrated, creating a space for an explicit algorithm description for the search procedure. = Quality Substantial experiments show that the new method is effective in synthesizing more programs in a shorter amount of time than the baseline DeepCoder method. However, while the experiments clearly show that the new contributions works better, they are not covering all interesting ablations (e.g., Do certain DSL instructions have more impact on the overall performance? Do the results depend on the chosen DSL?) = Significance Faster program synthesis methods have the potential to improve end-user programming experiences and could thus be very impactful. However, the picked DSL in DeepCoder (and hence here) seems fairly removed from actual usage scenarios, and as no further experiments to other domains are provided, it is unclear if the method would actually be practical.

Reviewer 3



This paper presents an approach to use neural networks to guide the search of programs in a DSL to perform program synthesis from few input-output examples. In particular, it learns two networks f and g to predict the next statement and which variables to drop from the memory, given the current state of the program together with the desired output. During training, an auxiliary task of predicting operators (with lambda functions) is used to improve the training. The environment state is represented as a set of fixed dimensional embeddings, which are then passed through a dense block and pooled to obtain example order-independent representation. This representation is then used to compute three predictions: a multiclass statement prediction f, a multi-label binary classification g, and an auxiliary task of predicting statement operators and the lambda functions. The search is performed using the CAB extension for beam search that iteratively weakens the pruning heuristics to perform new beam search runs. The technique is evaluated on the DeepCoder dataset and is shown to significantly outperform DeepCoder and baselines both in terms of synthesis time and also in terms of the lengths of the programs that can be synthesized. Overall, this paper presents a nice idea to use program states (dynamic environments) for predicting next functions. The technique is well-presented and a comprehensive evaluation is performed with many different ablation settings to evaluate the usefulness of different design choices. The results in comparison to previous techniques like DeepCoder are quite impressive and I also liked various ablation experiments. The paper is mostly well written and clear, but the model description parts can be presented more clearly. The evaluation is described in detail with reasonable design choices. The idea of using dynamic state information and variable dropping for learning to search in synthesis is novel. The technique is evaluated on toy-ish integer benchmarks (unlike RobustFill or Karel with real-world synthesis benchmarks), but still can be useful for making progress on hard synthesis tasks. Given the current progress in neural program synthesis approaches, DeepCoder is quite a weak baseline now. Note that DeepCoder trains only example embeddings and doesn’t train the search component, and relies on an external search. In particular, with systems like RobustFill, where one trains the search as well during training would be a better baseline method to compare here. Having said that the paper makes a key contribution here of using dynamic program states for making predictions, unlike RobustFill that only uses the syntactic partial programs. One way to perform such a comparison can be to train PCCoder with environment embeddings consisting of only (input-output) states without the additional intermediate variables. That would clarify precisely the contribution of having dynamic program states during the prediction process. The model description for dense blocks is also not described in great detail and I have several questions about the model’s design choices. Since the dense blocks seem to have a rather significant impact on performance (73% vs 83%), it would be good to understand the key properties of such a network compared to previous models used for synthesis. As the model is performing pooling over the environment state embeddings, is it the case that it can be provided an arbitrary number of input-output examples at test time? What is the size of the possible statements |S|? With increasing number of choices for operator arguments in a more complex DSL, this set can easily blow up. Why not use different networks for predicting functions and operands? It seems the auxiliary task of predicting functions is anyways being helpful. What happens if one adds other auxiliary task of predicting operands? It was also not clear what is the complexity of the programs that PCCoder can currently synthesize. In the author response, it would be great if the authors can provide a few samples of length 12 and length 14 programs that PCCoder can synthesize together with the corresponding input-output examples. In figure 4, what are the 8 different programs and the environments? For embedding program states, the current approach uses a relatively simple concatenation of variable embeddings. It might be interesting to explore other ways to embed program states such as the ones suggested in [1]. 1. Dynamic Neural Program Embedding for Program Repair. Ke Wang, Rishabh Singh, Zhendong Su. ICLR 2018